# Anti-Inflammatory Action of Dexmedetomidine on Human Microglial Cells

**DOI:** 10.3390/ijms231710096

**Published:** 2022-09-03

**Authors:** Sho Yamazaki, Keisuke Yamaguchi, Akimasa Someya, Isao Nagaoka, Masakazu Hayashida

**Affiliations:** 1Department of Anesthesiology and Pain Medicine, Juntendo University Graduate School of Medicine, 2-1-1 Hongo, Bunkyo-Ku, Tokyo 113-8421, Japan; 2Department of Anesthesiology and Pain Medicine, Juntendo Tokyo Koto Geriatric Medical Center, 3-3-20 Shinsuna, Koto-Ku, Tokyo 136-0075, Japan; 3Department of Biochemistry and Systems Biomedicine, Juntendo University Graduate School of Medicine, 2-1-1 Hongo, Bunkyo-Ku, Tokyo 113-8421, Japan; 4Faculty of Medical Science, Juntendo University, 6-8-1 Hinode, Urayasu 279-0013, Japan

**Keywords:** dexmedetomidine, microglia, anti-inflammatory action, delirium, signaling molecule, cytokine

## Abstract

Neuroinflammation, where inflammatory cytokines are produced in excess, contributes to the pathogenesis of delirium. Microglial cells play a central role in neuroinflammation by producing and releasing inflammatory cytokines in response to infection, tissue damage and neurodegeneration. Dexmedetomidine (DEX) is a sedative, which reduces the incidence of delirium. Thus, we hypothesized that DEX may alleviate delirium by exhibiting anti-inflammatory action on microglia. In the present study, we investigated the anti-inflammatory action of DEX on human microglial HMC3 cells. The results indicated that DEX partially suppressed the IL-6 and IL-8 production by lipopolysaccharide (LPS)-stimulated HMC3 cells as well as the phosphorylation of p38 MAPK and IκB and the translocation of NF-κB. Furthermore, DEX substantially suppressed IL-6 and IL-8 production by unstimulated HMC3 cells as wells as the phosphorylation of p38 MAPK and IκB and the translocation of NF-κB. These observations suggest that DEX exhibits anti-inflammatory action on not only LPS-stimulated but also unstimulated microglial cells via the suppression of inflammatory signaling and cytokine production.

## 1. Introduction

Perioperative disturbances of cognition occur acutely in the form of postoperative delirium (POD) as postoperative cognitive dysfunction (POCD). Delirium is defined as a fluctuating disturbance of consciousness with reduced ability to focus, sustain, or shift attention, accompanied by a change in cognition and perceptual disturbances. Delirium is the most frequent neuropsychiatric syndrome in the hospital, especially in older patients with pre-existent cognitive impairment. Delirium is associated with impaired physical and cognitive recovery, increased hospital costs and a higher mortality, and neuroinflammation is involved in the pathogenesis of delirium. The patients who develop delirium have a significantly longer duration of mechanical ventilation and longer intensive care unit (ICU) length of stay, as well as an over twofold higher risk of mortality. Interestingly, it has been reported that the high blood level of IL-6 is observed in patients with delirium during hospitalization in the ICU [1]. Therefore, controlling neuroinflammation is therapeutically important for limiting the risk of developing POD/POCD.

Microglia are located in the central nervous system (CNS) and interact with neural synapses. Microglia also play an important role in the innate immune system and neuroinflammation of CNS [2]. In response to injury or inflammatory stimuli, microglial cells are rapidly activated and promote neuroinflammatory reactions through the secretion of various chemokines and cytokines. Increased activation of microglia is also evident in POD. Interestingly, increased levels of inflammatory mediators (such as IL-6) are known to have a high incidence of POD [3].

Dexmedetomidine (DEX) is a sedative with the action of an alpha 2-adrenergic receptor agonist primarily used in the ICU. Recent studies have confirmed that DEX exerts the protective actions on various organ injuries [4]. Interestingly, clinical studies have revealed that DEX lowers the incidence of delirium in ICU compared to other sedatives [5]. Moreover, an animal model study indicated that DEX reduces the lipopolysaccharide (LPS)-induced neuroinflammation in the mouse brain and the cytokine-related changes in the disease behavior [6]. DEX also reduces the severity of lidocaine-induced spinal cord injury in rats by suppressing the expression of pro-inflammatory cytokines IL-1β and IL-18 [7].

In the present study, we examined the effects of DEX on LPS-induced inflammatory reactions by human microglial HMC3 cells, focusing on the activation of signaling molecules (p38 MAPK, ERK1/2 and IκB) and the nuclear translocation of NF-κB.

## 2. Results

### 2.1. Effect of LPS and DEX on the Viability of HMC3 Cells

HMC3 cells were incubated with or without 1000 nM DEX for 30 min, and then stimulated with or without 100 ng/mL LPS for 24 h. Thereafter, the LDH activity of the culture supernatants was assessed. The incubation of HMC3 cells with LPS and/or DEX did not affect the viability of HMC3 cells assessed by the LDH activity compared to that without LPS and DEX.

### 2.2. Effect of DEX on LPS-Induced p38 MAPK and ERK1/2 Activation

First, we evaluated the effect of DEX on the phosphorylation of p38 MAPK. HMC3 cells were incubated with or without 1000 nM DEX for 30 min and then stimulated with or without 100 ng/mL LPS for 15 min. The stimulation of HMC3 cells with LPS slightly increased the phosphorylation of p38 MAPK, and DEX marginally suppressed the LPS-induced phosphorylation of p38 MAPK. Interestingly, DEX partially suppressed the phosphorylation of p38 MAPK in unstimulated HMC3 cells (Figure 1A).

Next, we evaluated the effect of DEX on the phosphorylation of ERK1/2. In contrast to p38 MAPK, the stimulation of HMC3 cells with LPS (100 ng/mL) did not essentially increase the phosphorylation of ERK1/2, and DEX (1000 nM) did not affect the phosphorylation of ERK1/2 in LPS-stimulated and unstimulated HMC3 cells (Figure 1B).

### 2.3. Effect of DEX on LPS-Induced IκB Activation and Translocation of NF-κB 

Furthermore, we evaluated the effect of DEX on phosphorylation of IκB. HMC3 cells were incubated with or without 1000 nM DEX for 30 min and then stimulated with or without 100 ng/mL LPS for 15 min. The stimulation of HMC3 cells with LPS significantly increased the phosphorylation of IκB (*p* < 0.05), and DEX suppressed the LPS-induced phosphorylation of IκB (*p* < 0.05). Interestingly, DEX significantly suppressed the phosphorylation of IκB in unstimulated HMC3 cells (*p* < 0.05) (Figure 2A).

Moreover, we evaluated the effect of DEX on the nuclear translocation of NF-κB, a target molecule of IκB. Stimulation with LPS apparently increased the nuclear translocation of NF-κB, and DEX markedly suppressed the LPS-induced translocation of NF-κB. Importantly, DEX significantly suppressed the nuclear translocation of NF-κB in unstimulated HMC3 cells (*p* < 0.05) (Figure 2B).

### 2.4. Effect of DEX on LPS-Induced IL-6 and IL-8 Production and COX-2 Expression

Furthermore, we evaluated the effect of DEX on IL-6 and IL-8 production by LPS-stimulated HMC3 cells. HMC3 cells were incubated with or without 1000 nM DEX for 30 min, and then stimulated with or without 100 ng/mL LPS for 24 h. Stimulation of HMC3 cells with LPS significantly increased the production of IL-6 (*p* < 0.01) and IL-8 (*p* < 0.05), and DEX partially suppressed the LPS-induced the production of IL-6 and IL-8 (Figure 3A,B). Interestingly, DEX significantly suppressed the production of IL-6 and IL-8 by unstimulated HMC3 cells (*p* < 0.05).

Finally, we evaluated the effect of DEX on the COX-2 expression in LPS-stimulated HMC3 cells. Stimulation of HMC3 cells with LPS (100 ng/mL) apparently increased the expression of COX-2, and DEX substantially suppressed the LPS-induced the expression of COX-2. However, DEX did not affect the expression of COX-2 in unstimulated HMC3 cells (Figure 3C).

## 3. Discussion

There is increasing evidence that inflammation in the CNS contributes to the onset of neurodegenerative diseases and the progression of cognitive dysfunction. Importantly, neuroinflammation is associated with POD [8,9,10,11,12]. Microglia, the glial cells that support neurons, are thought to play a central role in neuroinflammation by producing and releasing inflammatory cytokines in response to infection, tissue damage, and neurodegeneration [13,14]. Furthermore, uncontrolled levels of microglial inflammation are involved in the pathogenesis of neurodegenerative diseases [15], and the overwhelming inflammation caused by infections or traumatic stimuli leads to neurodegeneration and nerve destruction [16]. DEX is a sedative with the action of an alpha 2-adrenergic receptor agonist and reduces the incidence of delirium in patients in ICU compared to other sedatives [17]. Moreover, some anti-inflammatory drugs and their efficiency on neural cells such as glial cells involving astrocytes were reported [18,19,20]. In the present study, we examined the effects of DEX on LPS-induced inflammatory reactions by human microglial HMC3 cells [21], focusing on the activation of signaling molecules (p38 MAPK, ERK1/2 and IκB) and the nuclear translocation of NF-κB.

Following the cell activation, IκB, an inhibitory molecule of NF-κB, is phosphorylated and degraded, and NF-κB translocates to the nucleus and promotes the expression of inflammatory mediators such as COX-2, cytokine and chemokines [22,23]. Mitogen-activated protein kinases (MAPKs) are a family of serine/threonine kinases with at least three major subfamilies, including p42/44 MAPKs (also called extracellular signal-regulated kinases: ERK-1 and ERK-2) and p38 MAPKs [11]. MAPKs are activated by nerve damage and inflammation. Interestingly, it has been reported that MAP/ERK signaling modulates the development of cognitive and emotional function and is involved in pathological and neurodegenerative processes [24]. Thus, in the present study, to elucidate the anti-inflammatory action of DEX, we examined the effects of DEX on LPS-induced inflammatory reactions by human microglial HMC3 cells, focusing on the activation of signaling molecules (p38 MAPK, ERK1/2 and IκB) and the nuclear translocation of NF-κB.

The present results indicate that LPS stimulation induced the phosphorylation of IκB and p38 MAPK but not ERK, and the translocation of NF-κB as well as the IL-6/-8 production and COX expression in HMC3 cells. These results suggest that IκB phosphorylation, NF-κB translocation and p38 MAPK phosphorylation, but not ERK phosphorylation, are involved in the IL-6/-8 production and COX expression by LPS-stimulated HMC3 cells. Furthermore, we evaluated the action of DEX on the LPS-stimulated HMC3 cells. The results indicated that DEX inhibits the LPS-induced phosphorylation of IκB and p38 MAPK, and translocation of NF-κB as well as the IL-6/-8 production and COX expression in HMC3 cells. Thus, DEX likely inhibits IL-6/-8 production and COX expression by LPS-stimulated HMC3 cells via the suppression of signaling mediated by IκB phosphorylation, NF-κB translocation and possibly p38 MAPK phosphorylation.

Interestingly, in unstimulated cells, DEX inhibits IκB phosphorylation, NF-κB translocation and p38 MAPK phosphorylation as well as IL-6/-8 production. These results suggest that DEX suppresses IL-6/-8 production in unstimulated cells via the inhibition of IκB phosphorylation, NF-κB translocation and p38 MAPK phosphorylation. In contrast, the IκB phosphorylation, NF-κB translocation and p38 MAPK phosphorylation are unlikely involved in the expression of COX-2 in unstimulated cells, since DEX suppressed the IκB phosphorylation, NF-κB translocation and p38 MAPK phosphorylation but not COX-2 expression. Thus, DEX is expected to suppress the low level of cytokine production by unstimulated microglial cells.

It has been reported that serum IL-6 and 8 levels were associated with delirium severity [25]. Moreover, a proinflammatory molecule COX-2 is also upregulated and induces prostaglandin E2 (PGE2), which promotes neuroinflammation and impairs memory function [26,27,28]. Interestingly, DEX inhibits LPS-induced production of PGE2 in microglia [29,30]. The present study has indicated that DEX, which lowers the incidence of delirium, suppresses inflammatory signals and suppresses the expression of inflammatory molecules not only in LPS-stimulated cells but also in unstimulated cells. Thus, it can be speculated that DEX may ameliorate neuroinflammation-mediated delirium by exhibiting anti-inflammatory action on microglial cells possibly via the suppression of inflammatory signaling and inflammatory molecule production.

The proposed model of the possible mechanism for the action of DEX against LPS-induced inflammatory response in human microglia HMC3, is shown in Figure 4.

In conclusion, our observations suggest that DEX exhibits the anti-inflammatory action on not only LPS-stimulated but also unstimulated microglial cells via the suppression of inflammatory signaling and cytokine production.

## 4. Materials and Methods

### 4.1. Materials

Human microglial HMC3 cell lines (CRL-3304) were obtained from the ATCC (Manassas, VA, USA); LPS was obtained from Sigma-Aldrich Co. (St. Louis, MO, USA). Minimal Essential Medium (MEM), non-essential amino acids (NEAA), sodium pyruvate and fetal bovine serum (FBS) were all obtained from Gibco BRL Life Technologies (Grand Island, NY, USA). In addition, phosphate-buffered saline (PBS), radio-immunoprecipitation assay (RIPA) buffer, sample buffer solution containing reducing reagent for SDS-PAGE (6×), running buffer solution for SDS-PAGE (10×), protease inhibitor cocktail, Blocking One, WB Stripping Solutions Strong and Protein Ladder One Multicolor (Broad Range) for SDS-PAGE were all purchased from Nacalai Tesque (Kyoto, Japan); ELISA kits of IL-6 and IL-8, BCA (bicinchoninic acid) protein assay reagent kit, enhanced chemiluminescent reagent (Super Signal West Dura), and NE-PER nuclear and cytoplasmic extraction reagents are purchased from Thermo Fisher Scientific (Waltham, MA, USA); Mini-PROTEAN^®^ TGX™ Precast Gel and Trans-Blot^®^, Turbo™ Mini PVDF Transfer Packs were purchased from Bio-Rad Laboratories (Hercules, CA, USA). LDH cytotoxicity detection kit was purchased from Takara Bio Inc. (Shiga, Japan).

### 4.2. Antibodies

The following antibodies were purchased from Cell Signaling Technology, Inc. (Danvers, MA, USA); rabbit anti-phospho-ERK1/2 MAPK (Thr202/Tyr204) antibody (#9101), rabbit anti-ERK1/2 MAPK antibody (#9102), rabbit monoclonal anti-phospho-NF-κB p65 (Ser536) antibody (#3033), rabbit monoclonal anti-NF-κB p65 antibody (#8242), rabbit anti-phospho-IκBα (S32) antibody (#2859), rabbit IκBα (44D4) antibody (#44D4), rabbit monoclonal anti-cyclooxygenase (COX) 2 (D5H5) XP antibody (#12282), and rabbit monoclonal histone H3 antibody (D1H2) XP (#4499). In addition, rabbit anti-phospho p38 MAPK antibody (Thr180/Tyr182) was obtained from Promega Corporation (Madison, WI, USA) and mouse monoclonal anti-p38 MAPK (p38/SAPK2) antibody (#612168) was obtained from BD Biosciences (San Jose, CA, USA). Mouse anti-α tubulin (p38/SAPK2) antibody (#612168) was obtained from Thermo Fisher Scientific (Waltham, MA, USA), Anti-glyceraldehyde-3-phosphate dehydrogenase (GAPDH) was obtained from Merck Millipore (Burlington, MA, USA). Horseradish peroxidase (HRP)-conjugated goat anti-rabbit IgG antibody (AP132P) and HRP-conjugated goat anti-mouse IgG/IgM antibody (AP308P) were obtained from Chemicon International (Temecula, CA, USA).

### 4.3. Cell Culture

HMC3 cells were incubated in MEM supplemented with 10% heat-inactivated FBS, 1% (*v*/*v*) penicillin/streptomycin, 1% (*v*/*v*) NEAA, and 1 mM sodium pyruvate. Cells were maintained at 37 °C in a humidified 5% CO_2_ atmosphere.

### 4.4. Cell Treatment and Cytotoxicity Assay

HMC3 cells were plated into 12-well tissue culture plates (1 × 10^5^ cells/well) and cultured in MEM with 10% FBS for 12 h. Cells were incubated with or without 1000 nM DEX for 30 min, and then stimulated with or without 100 ng/mL LPS for 24 h, based on the previously published protocol [31,32]. The culture supernatants were then collected and centrifuged at 12,000× *g* for 10 min, and LDH was measured using the LDH cytotoxicity detection kit according to the manufacturer’s instructions (Takara Bio Inc., Shiga, Japan) using an xMark™ microplate reader (Bio Rad, Hercules, CA, USA).

### 4.5. Preparation of Whole-Cell Lysates and Western Blots Analysis

HMC3 cells were plated into a 12-well tissue culture (1 × 10^5^ cells/well) and incubated in MEM containing 10% FBS for 12 h. Cells were incubated with or without 1000 nM DEX for 30 min, and then stimulated with or without 100 ng/mL LPS for 15 min (for p38 MAPK, ERK1/2 and IκB) or 24 h (for COX-2). Thereafter, the cells were washed three times with ice-cold PBS and lysed in 0.1 mL of RIPA buffer (50 mmol/l Tris-HCl pH 7.6, 150 mmol/l NaCl, 1% Nonidet P40, 0.5% sodium deoxycholate, 0.1% SDS) containing a protease inhibitor cocktail. The protein concentration of the cell lysates was measured using the BCA protein assay reagent. Lysates were mixed with sample buffer (62.5 mM Tris-HCl, pH 6.8, 2% SDS, 10% glycerol, 0.05% bromophenol blue, and 5% 2-mercaptoethanol) for sodium dodecyl sulfate-polyacrylamide gel electrophoresis (SDS-PAGE) and applied (10–20 μg of protein per lane) to 10% gels (Mini-PROTEAN^®^ TGX™ Precast Gel). Proteins were then electrotransferred onto polyvinylidene fluoride (PVDF) membranes (Trans-Blot^®^ Turbo™ Mini PVDF Transfer Packs). The membranes were incubated in Blocking One and probed with rabbit anti-phospho p38 MAPK antibody, anti-phospho ERK1/2 antibody, or anti-COX-2 antibody (all at 1:1000 dilution). The membranes were washed and then probed with HRP-conjugated goat anti-rabbit IgG at 1:10,000 dilution. Detection was performed using a Super Signal West Dura chemiluminescence substrate, and the signal was quantified using a LAS-3000 luminescence image analyzer (Fujifilm, Tokyo, Japan) and multi-gauge software (Fujifilm). Furthermore, the membranes were stripped using WB stripping solution Strong at 37 °C for 15 min. After washing, the membranes were proved with mouse anti-p38 MAPK antibody, rabbit anti-ERK1/2 antibody, rabbit anti-NF-κB antibody, or rabbit anti-tubulin (all diluted receptor1:1000), and further probed with HRP-labeled goat anti-mouse IgG/IgM or HRP-labeled goat anti-rabbit IgG at 1:10,000. Finally, the signals were visualized and analyzed.

### 4.6. Preparation of Nuclear Extract and Western Blot Analysis

HMC3 cells were plated into 6-well tissue culture plates (5 × 10^5^ cells/well) and incubated in MEM supplemented with 10% FBS for 12 h. Cells were incubated with or without 1000 nM DEX for 30 min and then stimulated with or without 100 ng/mL LPS for 15 min. Thereafter, the cells were washed three times with ice-cold PBS and detached by trypsinization, collected in the medium, and subjected to centrifugation at 300× *g* for 5 min. Nuclear extract was prepared using NE-PER nuclear and cytoplasmic extraction reagents kit, according to the manufacturer’s instructions. The cell pellet was suspended in hypotonic buffer and was centrifuged at 14,000× *g* for 30 min. After the collection of the supernatant (cytoplasmic fraction), the pellet (nuclear fraction) was lysed and solubilized in a lysis buffer containing a protease inhibitor cocktail (nuclear extract). The concentration was measured with BCA protein assay reagent. Nuclear extract was mixed with sample buffer for SDS-PAGE and applied to a 10% gel, followed by Western blotting with a rabbit anti-NF-κB antibody or rabbit anti-histone H3 antibody (both at 1:1000 dilution) and HRP-conjugated goat anti-rabbit IgG (1:10,000 dilution), as described above.

### 4.7. Quantification of IL-6 and IL-8

HMC3 cells were plated into 12-well tissue culture plates (1 × 10^5^ cells/well) and incubated in MEM supplemented with 10% FBS for 12 h. Cells were incubated with or without 1000 nM DEX for 30 min, and then stimulated with or without 100 ng/mL LPS for 24 h. The culture supernatants were then collected and centrifuged at 12,000× *g* for 10 min, and IL-6 and IL-8 were measured using the ELISA kits according to the manufacturer’s instructions (Thermo Fisher Scientific, Waltham, MA, USA) using an xMark™ microplate reader (Bio-Rad, Hercules, CA, USA).

### 4.8. Statistical Analysis

The results are presented as mean ± SD. The significance of differences was determined by multiple comparison tests and one-way ANOVA using the Tukey–Kramer method for post hoc testing using GraphPad Prism version 6.0 for Windows (GraphPad Software, San Diego, CA, USA). Statistical significance was found at *p* < 0.05.

## Figures and Tables

**Figure 1 ijms-23-10096-f001:**
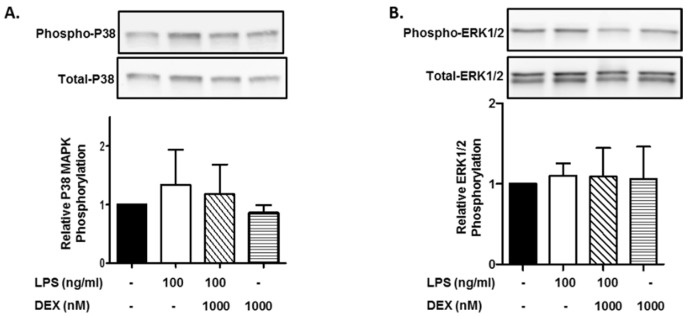
Effect of DEX on LPS-induced p38 MAPK and ERK1/2 activation. HMC3 cells were incubated with 1000 nM DEX for 30 min and further incubated with or without 100 ng/mL LPS for 15 min. Thereafter, the phosphorylation of p38 MAPK (**A**) and ERK1/2 (**B**) was analyzed by Western blotting. Phosphorylated p38 MAPK and ERK1/2 was normalized with total p38 MAPK and ERK1/2 and is expressed as a ratio to that of LPS-stimulated HMC3 cells without DEX. Representative images of Western blotting are shown above the graphs. Data are the means ± SD of 3 separate experiments.

**Figure 2 ijms-23-10096-f002:**
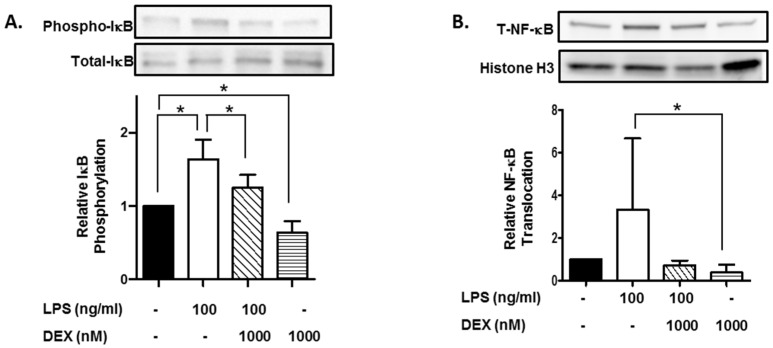
Effect of DEX on LPS-induced IκB activation and translocation of NF-κB. HMC3 cells were incubated with 1000 nM DEX for 30 min and further incubated with or without 100 ng/mL LPS for 15 min. Thereafter, the phosphorylation of IκB (**A**) and translocation of NF-κB (**B**) were analyzed by Western blotting. Phosphorylated IκB was normalized with total IκB and expressed as a ratio to that of LPS-stimulated HMC3 cells without DEX (**A**). Translocation of NF-κB was normalized with histone H3, and expressed as a ratio to that of LPS-stimulated HMC3 cells without DEX (**B**). Representative images of Western blotting are shown, above the graphs. Data are the means ± SD of 3 separate experiments. Data are compared between without and with LPS in the absence of DEX, between without and with DEX in the absence of LPS, and between without and with DEX in the presence of LPS. * *p* < 0.05.

**Figure 3 ijms-23-10096-f003:**
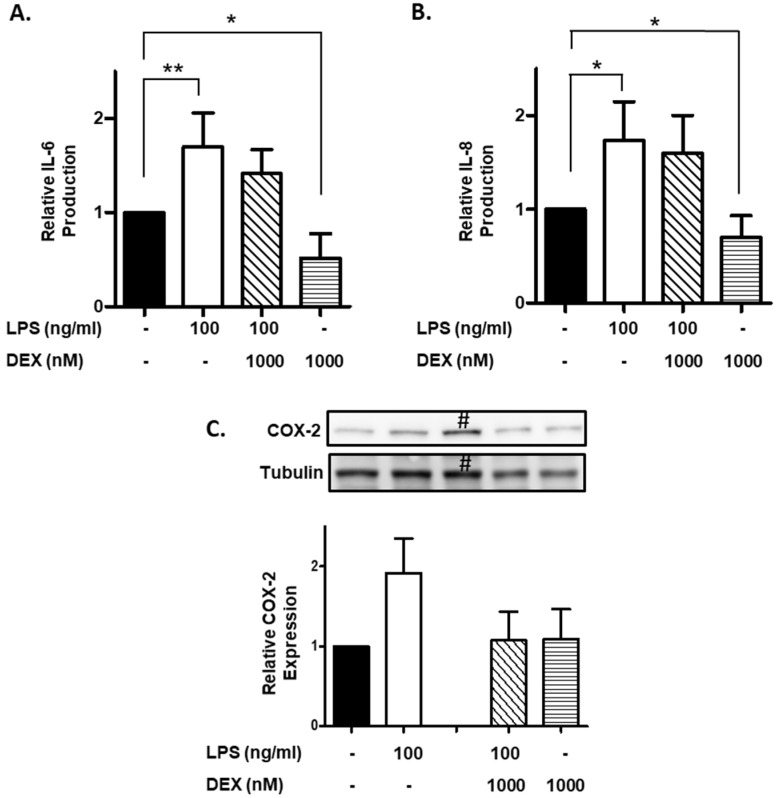
Effect of DEX on LPS-induced IL-6 and IL-8 production and COX-2 expression. HMC3 cells were incubated with 1000 nM DEX for 30 min, and then stimulated with 100 ng/mL LPS for 24 h. The culture supernatants were collected. The production of IL-6 (**A**) and IL-8 (**B**) was measured by ELISA and expressed as a ratio to that of resting HMC3 cells without LPS and DEX. Data represent the mean ± SD of six experiments. Data are compared between without and with LPS in the absence of DEX, and between without and with DEX in the absence of LPS. * *p* < 0.05, ** *p* < 0.001. After the incubation as described above, the cells were lysed in RIPA buffer containing a protease inhibitor cocktail, and expression of COX-2 was analyzed by Western blotting. The expression of COX-2 was normalized with tubulin and expressed as a ratio to that of LPS-stimulated HMC3 cells without DEX. Data are the means ± SD of 3 separate experiments. #: HMC3 cells incubated with LPS (100 ng/mL) and DEX (100 nM), which was excluded from calculation of the data (graph) shown in (**C**).

**Figure 4 ijms-23-10096-f004:**
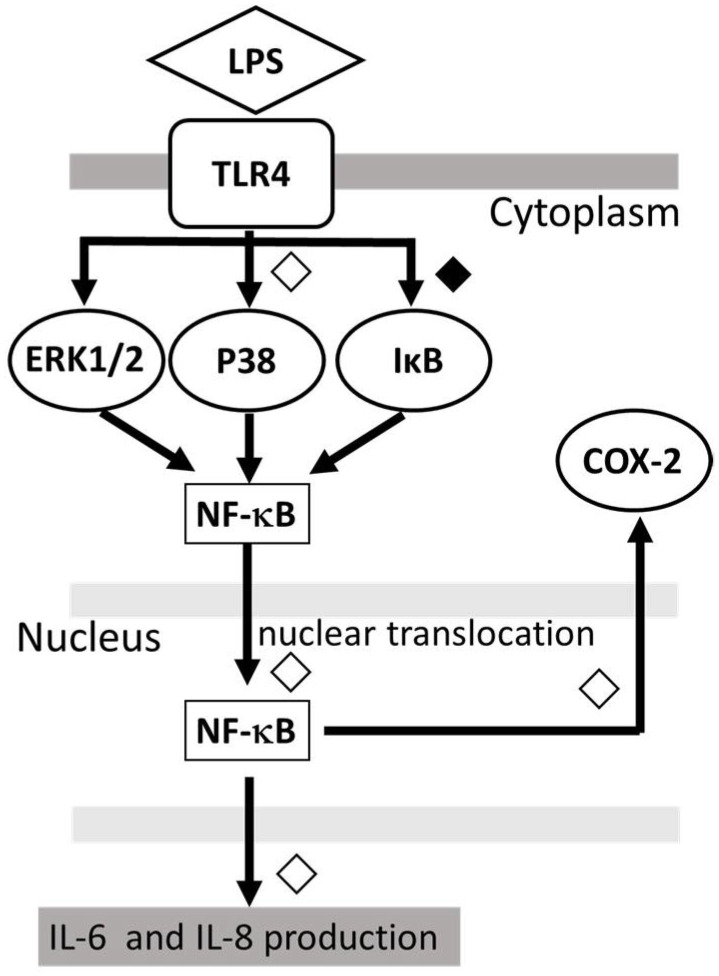
Proposed model showing that DEX modulates the LPS-stimulated inflammatory response. DEX suppressed the IL-6 and IL-8 production by LPS-stimulated HMC3 cells as well as the phosphorylation of p38 MAPK and IκB and the nuclear translocation of NF-κB. ◇; partial inhibition: ◆; significant inhibition.

## Data Availability

Not applicable.

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
