# Peer review of "Anti-Inflammatory Action of Dexmedetomidine on Human Microglial Cells"

_ijms, 2022, doi:10.3390/ijms231710096_

Round 1
Reviewer 1 Report
Yamaguchi et. al reported work "Anti-inflammatory Action of Dexmedetomidine on Human Microglial Cells" is interesting and well written. However it needs minor revision for the publication in IJMS.
1) There are several references related Anti-inflammatory Action of Dexmedetomidine, however in the current work authors reported its activity in Microglial cells, to inhibit the neuro inflammations. Therefore some of the recent references related neural cell anti-inflammatory drugs and their efficiency need to be briefly documented in Table by highlighting the current works significance.
2) Authors demonstrated several experiments IL-6 and IL-8 production inhibition, however plausible mechanism pathway need to be presented in figure.
3) There are few typo's in the manuscript, that to be corrected.
Author Response
Reviewer 1:
Query 1
There are several references related Anti-inflammatory Action of Dexmedetomidine, however in the current work authors reported its activity in Microglial cells, to inhibit the neuro inflammations. Therefore, some of the recent references related neural cell anti-inflammatory drugs and their efficiency need to be briefly documented in Table by highlighting the current works significance.
Response 1
Based on the comment of the reviewer, we have added the references related to the anti-inflammatory drugs and their efficiency on neural cells (such as astrocytes) (Refs. 18, 19, and 20) and explained in the Discussion section of the revised manuscript (lines 168; page 5). However, we did not explain the references using Table, because this paper is an original article, and the explanation of the references using Table is uncommon in an original article.
Query. 2
Authors demonstrated several experiments IL-6 and IL-8 production inhibition, however plausible mechanism pathway need to be presented in Figure. 4.
Response. 2
Based on the comment, we have added the figure showing the inhibitory mechanism of dexmedetomidine on IL-6 and IL-8 production as Figure 4 in the revised manuscript.
Query. 3
There are a few typos in the manuscript, that to be corrected.
Response. 3
Based on the comment, we have corrected some typos in manuscript.

Reviewer 2 Report
Anti-inflammatory Action of Dexmedetomidine on Human Microglial Cells.
Overall Review Comments
The authors attempted to study the anti-inflammatory action of dexmedetomidine (DEX), a selective α2-adrenergic receptor agonist. DEX has long been used successfully in several clinical settings for short-term sedation and analgesic sparing effects, and to reduce delirium in the intensive care unit in particular. This study mainly focuses on the causes of reduced incidence of delirium by exhibiting anti-inflammatory action on microglia. DEX partially suppressed multifunctional IL-6 and IL-8 production by lipopolysaccharide (LPS)-stimulated HMC3 cells and phosphorylation of p38MAPK and IkB and translocation of NF-kB. Preliminary observations concluded that DEX exhibits anti-inflammatory action on microglial cells via suppression of inflammatory signaling and cytokine production.
There have been significant gaps in the current understanding of the fates of microglia, astrocytes, and neurons during the episode of acute inflammatory activation. Although the cytokine dysregulations are associated to cause neuronal injury through a variety of cell specific signaling mechanisms, altered neurotransmission, activation of microglia and astrocytes could also lead to production of other signaling molecules such as free radicals, complement factors, glutamate, and nitric oxide etc. Elevated levels of chemokines from astrocytes have also been reported to be associated with delirium. The measurements of present experiments ignored to analyze the roles of other inflammatory cytokines, chemokines and inflammasomes. Exacerbation of neuroinflammation to produce deleterious outcomes such as delirium and associated disease progression (e.g., POCD) has not been sufficiently discussed in this study. As such, the preliminary in vitro results from the present experimental model do not make a significant contribution to current knowledge.
Other comments to authors
1) The present study uses 1mM DEX for 30 mins and 100ng/ml LPS for 24 h to activate and study the HMC3 cells. How are these functional dose/concentration and duration was determined?
2) How was the study determined to selectively choose to study the phosphorylation status of p38 MAPK and ERK1/2 and COX-2 expression?
Author Response
Reviewer 2:
Overall Review Comments
The authors attempted to study the anti-inflammatory action of dexmedetomidine (DEX), a selective α2-adrenergic receptor agonist. DEX has long been used successfully in several clinical settings for short-term sedation and analgesic sparing effects, and to reduce delirium in the intensive care unit in particular. This study mainly focuses on the causes of reduced incidence of delirium by exhibiting anti-inflammatory action on microglia. DEX partially suppressed multifunctional IL-6 and IL-8 production by lipopolysaccharide (LPS)-stimulated HMC3 cells and phosphorylation of p38MAPK and IkB and translocation of NF-kB. Preliminary observations concluded that DEX exhibits anti-inflammatory action on microglial cells via suppression of inflammatory signaling and cytokine production.
There have been significant gaps in the current understanding of the fates of microglia, astrocytes, and neurons during the episode of acute inflammatory activation. Although the cytokine dysregulations are associated to cause neuronal injury through a variety of cell specific signaling mechanisms, altered neurotransmission, activation of microglia and astrocytes could also lead to production of other signaling molecules such as free radicals, complement factors, glutamate, and nitric oxide etc. Elevated levels of chemokines from astrocytes have also been reported to be associated with delirium. The measurements of present experiments ignored to analyze the roles of other inflammatory cytokines, chemokines and inflammasomes. Exacerbation of neuroinflammation to produce deleterious outcomes such as delirium and associated disease progression (e.g., POCD) has not been sufficiently discussed in this study. As such, the preliminary in vitro results from the present experimental model do not make a significant contribution to current knowledge.
Response. 1
Thank you for your valuable suggestion. The paper presents only limited evidence regarding the effects of DEX on LPS-induced HMC3 cell line activation. In fact, the expression of another cytokine and chemokine such as TNF-α, IL-1b and iNOS has not been addressed. In the preliminary experiments, we examined the effect of LPS stimulation on TNF-α and IL-1b production by HMC3 cells. However, TNF-α and IL-1b production could not be induced by LPS under our experimental condition. Thus, in the present study, the effect of DEX was presented against the IL-6 and IL-8 production and COX-2 expression, which could be induced by LPS stimulation (as described in the Response to The Query 2 of the Reviewer 3). Based on your comment, the sentence “DEX may alleviate delirium by exhibiting anti-inflammatory action on microglia” has been deleted from the Introduction and Discussion (lines 60-61 , page 2 ; lines 184-185 , page 5 of the original manuscript), to emphasize our hypothesis in the conclusion of the Discussion “Thus, it can be speculated that DEX may ameliorate neuroinflammation-mediated delirium by exhibiting anti-inflammatory action on microglial cells possibly via the suppression of inflammatory signaling and inflammatory molecule production.”
Other comments to authors
Query. 1
The present study uses 1mM DEX for 30 mins and 100 ng/ml LPS for 24 h to activate and study the HMC3 cells. How are these functional dose/concentration and duration was determined?
Response. 1
We determined the functional dose/concentration and duration of LPS and DEX, based on the results of Shi et al (Molecules 2019, 24, 367; doi:10.3390/molecules24020367; Ref. 31) and Zhou et al. (Kaoshing J Med Sci 2019;35:750-756; Ref. 32), respectively. These are explained in the Materials and Methods section of the revised manuscrpt (lines 266-267; page 7).
Query. 2
How was the study determined to selectively choose to study the phosphorylation status of p38 MAPK and ERK1/2 and COX-2 expression?
Response. 2
p38 MAPK,ERK1/2, JNK and NF-κB are reported to be involved in the production of IL-6 and IL-8, as signaling molecules (Refs. 21). Thus, in the present study, we investigated the phosphorylation of p38 MAPK, ERK1/2 and JNK, and translocation of NF-κB. LPS stimulation induced the phosphorylation of p38 MAPK and ERK1/2, and the translocation of NF-κB but not the phosphorylation of JNK. Thus, the results of the phosphorylation of JNK is excluded from the manuscript. Moreover, the suppressive effect of DEX on COX-2 expression has been reported in the Ref. 30 using LPS-stimulated RAW264.7 mouse macrophage-like cells: Thus, the effect of DEX on COX-2 expression is investigated in the present study.

Reviewer 3 Report
The manuscript by Yamazaki et al. addresses mechanisms of action of dexmedetomidine (DEX) in a microglial cell line HMC3, by focusing on anti-inflammatory effects of the drug. DEX is used as a postoperative sedative to prevent the development of postoperative delirium (POD) and it has been described as alpha-adrenergic agonist. The authors present a concise set of 3 figures showing that the drug slightly reduces the production of 2 cytokines and which correlates with detectable effects on JNK and IkB phosphorylation and Cox2 expression, but not on activation of Erk. The topic is important, but the correlation between the effects of DEX on delirium and on LPS-induced inflammation is only speculative. In addition, there are three major points of criticism that must be addressed before any reevaluation of the work:
1. First criticism is a very limited novelty of the findings presented in this paper. The only element of novelty is the use of different cell line, with respect to previously published reports. Peng et al. (J. Surg. Res, 2013) showed suppression of IL1B, TNFalpha and iNOS expression in in primary microglial cells .Yeh et al. (Plos One, 2018), showed anti-inflammatory effects of DEX both, in vivo, in LPS-treated mice as well as in LPS-treated BV-2 microglial cells. Reduction of the pro-inflammatory cytokines IL-1β and IL-18 expression, attributed to the attenuation of microglial activation was also reported in lidocaine-induced neurotoxicity in rats (Ding et al. 2021, Biological and Pharmaceutical Bulletin). Anti-inflammatory activity and suppression of cytokine expression by DEX has been also reported several non-microglial cell lines (macrophages, ovarian cancer cells, lymphocytes etc..). This vast literature is only marginally discussed in the paper.
2. The paper presents only limited evidence regarding the effects of DEX on LPS-induced HMC3 cell line activation. In fact, the expression of TNF‐α, IL‐1β and iNOS has not been addressed, neither any possible effects on microglial polarization versus M1 or M2-like phenotypes.
3. There are important problems with the presented data: A) The loading controls in Fig. 2A and the blot presenting H3 levels in Fig. 2 B are from different gels. B) The quantification of the Western blot showing COX2 levels and tubulin levels in Fig. 3C could not give the results presented in the graph. In addition the blot has 5 lanes, while the OD quantification graph only 4 bars.
Several minor issues can be pointed out. For example:
1. the extensive, repetitive reporting of Western blotting procedures in figure legends, even though sufficient details are given in material and methods.
2. The outline of graphs is sloppy (fonts, alignments etc..).
3. In the conclusions of the discussion the authors write: “Thus, it can be speculated that DEX may ameliorate delirium by exhibiting anti-inflammatory action on microglial cells possibly via the suppression of inflammatory signaling and inflammatory molecule production.” This conclusion is unfortunate. By this statement the authors emphasize a limited value of their findings with respect to delirium, since the role of microglia and anti-inflammatory effects of DEX in delirium has been suggested several times before without the data presented in this manuscript.
Author Response
Reviewer 3:
Major comments
Query. 1
First criticism is a very limited novelty of the findings presented in this paper. The only element of novelty is the use of different cell line, with respect to previously published reports. Peng et al. (J. Surg. Res, 2013) showed suppression of IL-1B, TNF-alpha and iNOS expression in in primary microglial cells. Yeh et al. (Plos One, 2018), showed anti-inflammatory effects of DEX both, in vivo, in LPS-treated mice as well as in LPS-treated BV-2 microglial cells. Reduction of the pro-inflammatory cytokines IL-1β and IL-18 expression, attributed to the attenuation of microglial activation was also reported in lidocaine-induced neurotoxicity in rats (Ding et al. 2021, Biological and Pharmaceutical Bulletin). Anti-inflammatory activity and suppression of cytokine expression by DEX has been also reported several non-microglial cell lines (macrophages, ovarian cancer cells, lymphocytes etc.). This vast literature is only marginally discussed in the paper.
Response. 1
Thank you for your valuable suggestion. Based on the comments, we added the suggested reference, and explained the effect of DEX on not only LPS-stimulated but also lidocaine-induced neurotoxicity in the Introduction section of the revised manuscrpt (lines 59-61; page 2).
Query. 2
The paper presents only limited evidence regarding the effects of DEX on LPS-induced HMC3 cell line activation. In fact, the expression of TNF-α, IL-1b and iNOS has not been addressed, neither any possible effects on microglial polarization versus M1 or M2-like phenotypes.
Response. 2
In the preliminary experiments, we examined the effect of LPS stimulation on TNF-α and IL-1b production by HMC3 cells. However, TNF-α and IL-1b production could not be induced by LPS under our experimental condition. Thus, in the present study, the effect of DEX was presented against the IL-6 and IL-8 production and COX-2 expression, which could be induced by LPS stimulation.
Query. 3
There are important problems with the presented data: A) The loading controls in Fig. 2A and the blot presenting H3 levels in Fig. 2 B are from different gels. B) The quantification of the Western blot showing COX2 levels and tubulin levels in Fig. 3C could not give the results presented in the graph. In addition the blot has 5 lanes, while the OD quantification graph only 4 bars.
Response. 3
As the figure legend, the lane which is marked with # was not counted in quantification.
Fig. 2A and Fig. 2B are the different results of western blotting using whole cell lysates and nuclear fractions, respectively. However, the blots of Phospho Iκ-B and Total Iκ-B (Fig. 2A), and Total NF-κB and Histone H3 (Fig. 2B) represent the same gels, respectively, because the membranes used for the detection of Phospho Iκ-B and Total NF-κB were stripped, and the same membranes were used for the detection of Total Iκ-B and Histone H3, respectively.
As pointed by the reviewer, in Fig. 3C, the blot has 5 lanes, while the OD quantification graph has only 4 bars. One of the signals (indicated by #) indicates the result of HMC3 cells incubated with LPS (100 ng/ml) and DEX (100 nM), and is not corresponded to the results of the graph shown in (C); thus, the signal was excluded from calculation of the data (graph) shown in (C), as explained in the last sentence of the legend for Figure 3.
Minor comments
Query. 1
The extensive, repetitive reporting of Western blotting procedures in figure legends, even though sufficient details are given in material and methods.
Response. 1
Based on the comment of the reviewer, the repetitive descriptions of Western blotting procedures in figure legends have been deleted in the revised manuscript.
Query. 2
The outline of graphs is sloppy (fonts, alignments etc.)
Response. 2
Based on the comment of the reviewer, the fonts and alignments of the figures have been corrected in the revised manuscript.
Query. 3
In the conclusions of the discussion the authors write: “Thus, it can be speculated that DEX may ameliorate delirium by exhibiting anti-inflammatory action on microglial cells possibly via the suppression of inflammatory signaling and inflammatory molecule production.” This conclusion is unfortunate. By this statement the authors emphasize a limited value of their findings with respect to delirium, since the role of microglia and anti-inflammatory effects of DEX in delirium has been suggested several times before without the data presented in this manuscript.
Response. 3
Based on the comment of the reviewer, the sentence “DEX may alleviate delirium by exhibiting anti-inflammatory action on microglia” has been deleted from the Introduction and Discussion (lines 60-61 , page 2 ; lines 184-185 , page 5 of the original manuscript), to emphasize our hypothesis in the conclusion of the Discussion “Thus, it can be speculated that DEX may ameliorate neuroinflammation-mediated delirium by exhibiting anti-inflammatory action on microglial cells possibly via the suppression of inflammatory signaling and inflammatory molecule production.”
